# JUMANJI: A DIVERSE SUITE OF SCALABLE REINFORCEMENT LEARNING ENVIRONMENTS IN JAX

**Clément Bonnet**[1]*     **Daniel Luo**[1]     **Donal Byrne**[1]     **Shikha Surana**[1]

**Sasha Abramowitz**[1]     **Paul Duckworth**[1]     **Vincent Coyette**[1]     **Laurence I. Midgley**[2]

**Elshadai Tegegn**[1]     **Tristan Kalloniatis**[1]     **Omayma Mahjoub**[1]     **Matthew Macfarlane**[3]

**Andries P. Smit**[1]     **Nathan Grinsztajn**[1]     **Raphael Boige**[1]     **Cemlyn N. Waters**[1]

**Mohamed A. Mimouni**[1]     **Ulrich A. Mbou Sob**[1]     **Ruan de Kock**[1]     **Siddarth Singh**[1]

**Daniel Furelos-Blanco**[4]     **Victor Le**[1]     **Arnu Pretorius**[1]     **Alexandre Laterre**[1]

[1]InstaDeep    [2]University of Cambridge    [3]University of Amsterdam    [4]Imperial College London

## ABSTRACT

Open-source reinforcement learning (RL) environments have played a crucial role in driving progress in the development of AI algorithms. In modern RL research, there is a need for simulated environments that are performant, scalable, and modular to enable their utilization in a wider range of potential real-world applications. Therefore, we present Jumanji, a suite of diverse RL environments specifically designed to be *fast*, *flexible*, and *scalable*. Jumanji provides a suite of environments focusing on combinatorial problems frequently encountered in industry, as well as challenging general decision-making tasks. By leveraging the efficiency of JAX and hardware accelerators like GPUs and TPUs, Jumanji enables rapid iteration of research ideas and large-scale experimentation, ultimately empowering more capable agents. Unlike existing RL environment suites, Jumanji is highly customizable, allowing users to tailor the initial state distribution and problem complexity to their needs. Furthermore, we provide actor-critic baselines for each environment, accompanied by preliminary findings on scaling and generalization scenarios. Jumanji aims to set a new standard for speed, adaptability, and scalability of RL environments.

## 1 INTRODUCTION

High-quality datasets and benchmarks are crucial to the development of AI research (Deng et al., 2009; Krizhevsky et al., 2012; Bellemare et al., 2013). They allow for coordinated research on problems that serve as a measure of progress toward shared goals. However, most currently open-sourced reinforcement learning (RL) environment libraries are not closely tied to practical problems. Furthermore, in industrial settings operating at scale, these libraries do not provide sufficient flexibility and scalability to facilitate long-term AI research suitably close to real-world applications.

For RL to be useful in the real world, further research progress is needed. This will require benchmarks that are: (1) *fast*, i.e. hardware-accelerated to overcome sequential bottlenecks and allow for faster experiment iteration; (2) *flexible*, by allowing easy customization to capture realistic problem settings

---

*Correspondence to: `<clement.bonnet16@gmail.com>`

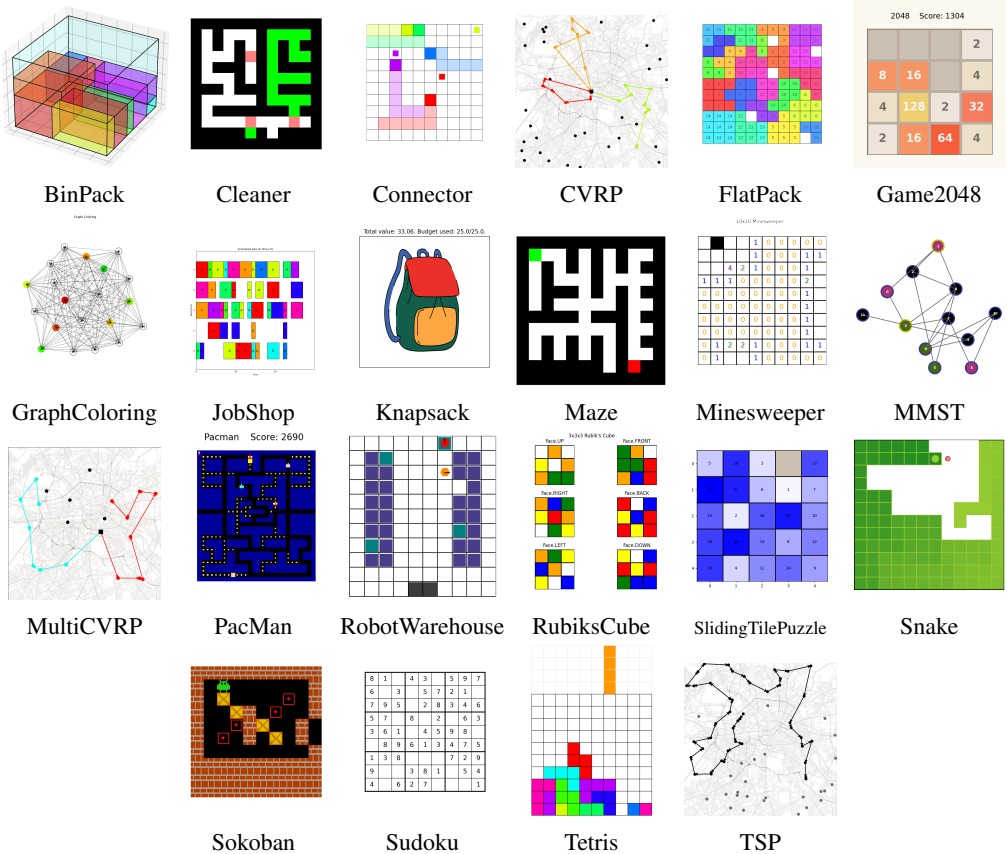

Figure 1: All 22 environments implemented in Jumanji (in alphabetic order) are divided into three categories. **Routing** problems: Cleaner, Connector CVRP (Capacitated Vehicle Routing Problem), Maze, MMST (Multiple Minimum Spanning Tree), MultiCVRP (Multiple-vehicle CVRP), PacMan, RobotWarehouse, Snake, Sokoban, and TSP (Travelling Salesman Problem). **Packing** problems: BinPack, FlatPack, JobShop, Knapsack, and Tetris. **Logic** games: Game2048, GraphColoring, Minesweeper, RubiksCube, SlidingTilePuzzle, and Sudoku.

of interest (e.g. intrinsic stochasticity and distribution shift); and (3) *scalable*, to be able to arbitrarily set the level of difficulty of an environment, ensuring a more faithful representation of the challenges that arise in real-world problems.

Although many other popular benchmarks exist such as the Arcade Learning Environment (ALE) (Bellemare et al., 2013), OpenAI Gym (Brockman et al., 2016), and Gymnax (Lange, 2022), which satisfy a subset of the above criteria, none of them have managed to cover all three. We believe it is crucial to fill this gap to help push RL research closer to industrial applications.

In this paper, we introduce **Jumanji**: a diverse suite of *fast*, *flexible*, and *scalable* RL environments. Jumanji is organized into three problem categories: routing, packing, and logic. At its core is a set of NP-hard combinatorial optimization problems (COPs) that closely resemble problems encountered in the real world. The environment dynamics of these problems seamlessly scale with complexity allowing for long-term research suitable for real-world industrial applications. Jumanji is written in JAX (Bradbury et al., 2018), to leverage composable transformations with automatic differentiation and the XLA compiler for highly efficient RL systems that run directly on GPU or TPU accelerators. Furthermore, Jumanji promotes flexibility by allowing arbitrary initial state distributions via easily modifiable reset functions and bespoke generators. We empirically demonstrate the capabilities of Jumanji through a set of initial experiments. Specifically, we present results on training an actor-critic agent across all environments, establishing a benchmark useful for future comparisons. We show that Jumanji environments are highly scalable, demonstrating high throughput in both a single-device and

multi-device setting. Finally, we illustrate the flexibility of environments by customizing initial state distributions to study generalization in a real-world problem setting.

The main contributions of this paper are as follows:

1. We introduce Jumanji: an open-source and diverse suite of industry-inspired RL environments, that are *fast*, *flexible*, and *scalable*.

2. We provide baseline actor-critic agents for all environments.

3. We present initial experiments demonstrating that, unlike existing RL benchmarks, Jumanji environments offer a high degree of scalability and flexibility.

## 2 RELATED WORK

Benchmark environments have been pivotal in the development and evaluation of RL algorithms. OpenAI Gym (Brockman et al., 2016), with its diverse task suite and user-friendly API, has become a benchmarking standard in the field. Other libraries, such as DMLab (Beattie et al., 2016) for complex 3D environments, and Mujoco (Todorov et al., 2012) for high-fidelity physics simulations, have enabled researchers to push the boundaries of agent capabilities. However, despite the significant contributions, these libraries have limitations in computational efficiency and scalability.

**Hardware-accelerated Environments**   A common approach to increasing environment throughput is through parallelization of the environment itself. Prior work such as EnvPool (Weng et al., 2022) utilizes multiple CPU cores and C++ based threading of multiple instances of an environment in order to expedite the bottleneck of sequential simulation steps. GPU-accelerated environments like Nvidia's CuLE (Dalton et al., 2019; Dalton & frosio, 2020) and Isaac Gym (Makoviychuk et al., 2021) take a different approach, leveraging the parallel processing capabilities of GPUs. CuLE provides a CUDA port of ALE (Bellemare et al., 2013), rendering frames directly on GPU, whilst Isaac Gym provides an accelerated alternative to Mujoco. Although these environments offer significant efficiency gains, they are strictly limited to GPUs and cannot be readily extended to other hardware accelerators (e.g. TPUs). JAX (Bradbury et al., 2018) is a numerical computing library that leverages automatic differentiation, vectorization, parallelization, and an XLA compiler for device-agnostic optimization. JAX is utilized in RL environments such as Brax (Freeman et al., 2021), a differentiable physics engine, Pgx (Koyamada et al., 2023), a collection of board game simulators, Gymnax (Lange, 2022), a library of popular RL environments re-implemented in JAX, JaxMARL (Rutherford et al., 2023), a collection of commonly used MARL environments, and Craftax-Classic (Matthews et al., 2024) a re-implementation in JAX of the open-ended environment Crafter. These environments represent a significant advance in efficiency but are limited in scope and flexibility.

**Combinatorial Optimization Problems (COPs)**   COPs present a significant area of research in RL, with many real-world and industrial applications. Examples include the Traveling Salesman Problem, Bin Packing, the Capacitated Vehicle Routing Problem, and the Knapsack Problem (Bengio et al., 2021). While there have been substantial advances in related software, such as Google OR Tools (Google, 2023), there is a noticeable gap in support for RL-based COP solutions. Libraries such as OR RL Benchmarks (Balaji et al., 2019) and OR-Gym (Hubbs et al., 2020) provide COP environments that adhere to the standard Gym API, however, they are restricted to run on CPU, making it difficult to parallelize and scale.

**Benchmark Diversity**   Multiple RL benchmarks have been proposed to facilitate agent generalization, such as Procgen (Cobbe et al., 2019), OpenSpiel (Lanctot et al., 2020), BSuite (Osband et al., 2020), and XLand (Team et al., 2021). Whilst providing challenging scenarios for training, Procgen and OpenSpiel do not inherently support scaling of the environment dynamics, and unlike Jumanji, none are designed to utilize hardware accelerators. Lastly, whilst no single benchmark suite can handle all situations, the ability to extend and create environments is crucial. Unity ML Agents (Juliani et al., 2020) is extendable and parallelizable via the Unity game engine, but not optimized for accelerators. Gymnasium aims to standardize Gym (Brockman et al., 2016) but does not directly provide a base for new environments. Most libraries rely on hard-coded components that require users to develop new extensions. Jumanji, however, follows a composition-based design,

allowing for easy modifications of initial state distributions, reset behaviors, level generators, and rendering.

## 3 JUMANJI

We introduce Jumanji (`v1.0.0`), a suite of 22 JAX-based environments, visualized in Figure 1. These diverse problems rely on a variety of geometries, including grids, graphs, and sets. The environments are organized into three problem categories: routing, packing, and logic. Many of these are NP-hard COPs inspired by real-world industry settings. Jumanji leverages JAX to significantly accelerate and parallelize simulation steps while remaining flexible and allowing for scalable problem complexity.

This section provides an overview of the library, first introducing the RL setting and then Jumanji's application programming interface (API).

### 3.1 RL PRELIMINARIES

Each Jumanji environment is structured as a Markov decision process (MDP) (Puterman, 1994), $\mathcal{M} = (S, A, \mu, P, R, \gamma)$, where $S$ is the state space, $A$ is the action space, $\mu$ is the initial state distribution, $P$ defines the environment transition dynamics, $R$ is the reward function, and $\gamma$ is the discount factor. We can generate trajectories from an MDP by rolling out the environment dynamics. That is, at time step $t$, an action $a_t$ transitions the environment from the current state $s_t$ to the next state $s_{t+1}$ as defined by the environment dynamics $P$, resulting in a reward $r_t$. The objective of an agent is often to maximize the discounted expected return, given by $\mathbb{E}_{a_t \sim \pi(\cdot|s_t)}[\sum_{t=0}^{T} \gamma^t r(s_t, a_t)]$, where $\gamma$ is the discount factor and $\pi$ is the agent's policy.

### 3.2 API

Jumanji's interface is lightweight, flexible, and capable of representing a diverse set of RL problems. It draws inspiration from OpenAI Gym (Brockman et al., 2016), dm_env (Muldal et al., 2019), and Brax (Freeman et al., 2021). It is flexible in three ways: (1) allowing customization of the initial state distribution via generators, (2) custom visualization via environment viewers, and (3) custom reward functions. Below, we introduce the key components of the Jumanji API, including the environment interface, state, observation, generators, specs, and registry.

**Environment Interface** The `Environment` interface defines the blueprint for Jumanji environments. Each environment must contain the following methods: `reset`, `step`, `observation_spec`, and `action_spec`. The API allows for optional `render` and `animate` methods to visualize a state or a sequence of states. For a code snippet demonstrating how to create a new environment by extending the environment interface, see Appendix A.4. Here, we provide code to instantiate an environment from the Jumanji registry, reset, step, and (optionally) render it:

```python
import jax
import jumanji

# Instantiate a Jumanji environment from the registry
env = jumanji.make('Snake-v1')

# Reset the environment
key = jax.random.PRNGKey(0)
state, timestep = jax.jit(env.reset)(key)

# Sample an action and take an environment step
action = env.action_spec().generate_value()
state, timestep = jax.jit(env.step)(state, action)

# (Optional) Render the environment state
env.render(state)
```

**State**   The `State` is a pytree (e.g. dataclass or namedtuple) that contains all the required information to transition the environment's dynamics for a given action. This is a design choice, and as such, Jumanji environments are stateless i.e. the `reset` and `step` methods are functionally pure. This allows Jumanji to leverage JAX's transformations (`jit`, `grad`, `vmap`, and `pmap`) to make the environments highly scalable. Every state includes a pseudorandom number generator (PRNG) key, which is used during a potential auto-reset and in the case of stochastic transitions.

**TimeStep and Observation**   The `TimeStep` contains all the information available to an agent about the state of the environment at a time step, namely; the `step_type` (first, mid, or last), `reward`, `discount`, `observation`, and `extras`. As such, it is based on the dm_env `TimeStep` but with an additional `extras` field, where environment metrics can be logged that are neither available to the agent nor part of the state. For Jumanji environments, the `Observation` is a JAX pytree, making it amenable to multiple data types.

**Generators**   For a given environment, a *generator* is used to define the initial state distribution. Jumanji provides flexibility by allowing the use of custom generators, enabling users to define an initial state distribution specific to their problem. In most environments, the `reset` method calls a generator that returns the initial state. The generator can be specified upon environment instantiation. The user can choose from a set of pre-existing generators or implement their own generator. If not specified, a default generator is used.

**Specs**   Inspired by dm_env, Jumanji specifications define the tree structure, shape, data type, and optionally the bounds of the observations, actions, rewards, and discounts. Jumanji's `Spec` class is more general than its dm_env counterpart, allowing for nested structures. This is achieved by implementing each spec as a nested JAX pytree composed of a set of primitive specs (`Array`, `BoundedArray`, `DiscreteArray`, or `MultiDiscreteArray`) which form the leaves of the tree while each non-leaf node is itself another nested `Spec` object. Environments in Jumanji have their action space described as a spec, which means although current environments have discrete actions, Jumanji supports both discrete and continuous action spaces.

**Registry**   Jumanji keeps a strict versioning of its environments for the purpose of reproducibility. This is achieved through a registry of standard environments with their respective configurations. For each environment, a version suffix is appended, e.g. "Snake-v1". When significant changes are made to environments, the version number is incremented.

## 4   JUMANJI BENCHMARK

In this section, we first describe a highly efficient method for training RL agents in Jumanji environments. Secondly, we provide details of standard actor-critic baseline agents. Finally, we present experiments demonstrating the speed and parallelization of Jumanji environments.

### 4.1   EFFICIENT TRAINING

As described in Section 3.2, Jumanji environments are designed to be stateless, allowing Jumanji to take full advantage of JAX's transformations. JAX-based stateless environments provide multiple benefits. Firstly, we can JIT-compile the full training loop of an agent. This often includes rolling out the environment to generate trajectories, and then separately updating the parameters of the agent based on that experience. We provide a throughput ablation in appendix D.2 to demonstrate the speed-up that arises when removing data transfer between host and device. Secondly, JAX's `grad` allows for efficient computation of gradients using automatic differentiation. Thirdly, JAX's vectorization (`vmap`) can be used to generate rollouts and compute parameter updates in parallel on a single device. Finally, JAX's process parallelism (`pmap`) can be leveraged to parallelize the computations across multiple devices, where gradients are aggregated across devices using `pmean`. Hessel et al. (2021) propose the *Anakin* architecture for exactly this setting, with an emphasis on maximizing the utilization of TPU pods, although their approach is general and also applies to multi-GPU RL training. All the experiments in this paper are implemented using this efficient Anakin framework.

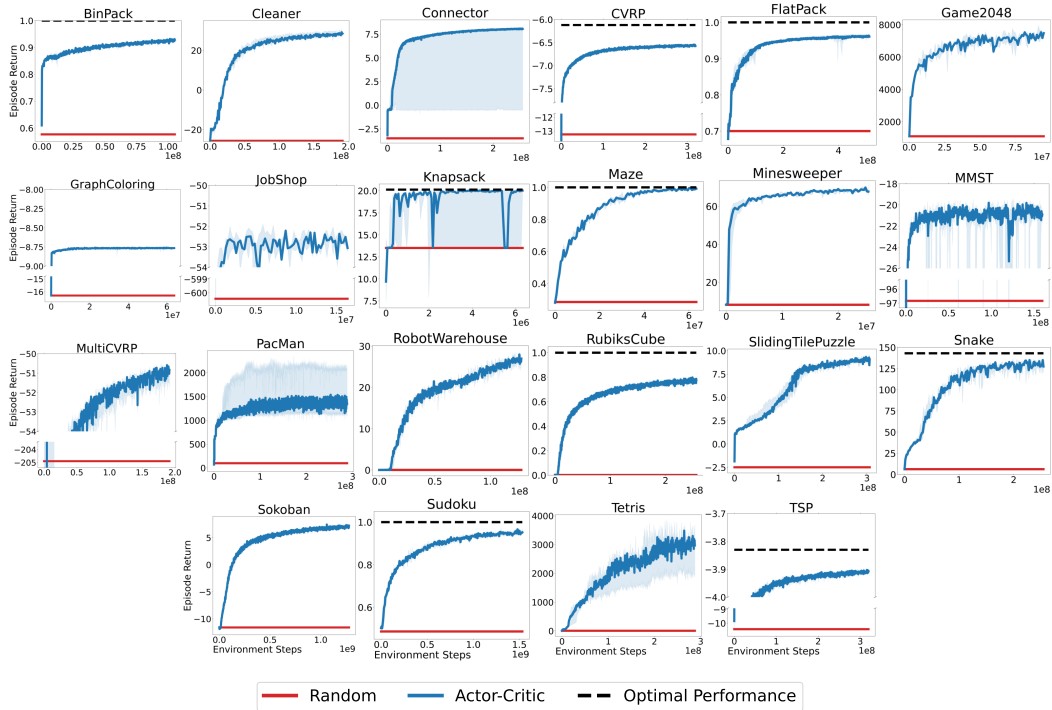

Figure 2: Learning curves from training an actor-critic agent (A2C) in blue compared to a masked random policy in red on all 22 Jumanji environments. When possible to determine, the optimal expected return is shown with a dashed line. Experiments were run with three different seeds, with the median represented as a blue curve and the min/max as the shaded region.

## 4.2 ACTOR-CRITIC BASELINE

Jumanji provides an implementation of an A2C (Mnih et al., 2016) agent, built using the DeepMind JAX ecosystem (Babuschkin et al., 2020). Since Jumanji environments use different geometries (e.g. images, sets, etc.), the agent relies on environment-specific neural networks, e.g. image inputs may be fed to a convolution neural network while permutation-equivariant problems may use a transformer architecture (Vaswani et al., 2017). To promote research using Jumanji, we open-source the algorithm, the training pipeline, checkpoints, and the aforementioned actor-critic networks which are compatible with any algorithms relying on a policy or state-value function. Appendix B provides further details on these environment-specific networks.

To benchmark Jumanji environments, we provide learning curves of our A2C implementation on all 22 environments. We compare our algorithm to the optimal performance (where possible to determine) and a random policy that uniformly samples actions from the set of valid actions. Note, that the optimal performances for TSP, CVRP, and Knapsack are taken from (Kwon et al., 2020). The experiments were performed on a TPUv3-8 using the Anakin framework. We refer the reader to Appendix C.1 for more details on the training. In Figure 2, we show the learning curves of the A2C agent on each of the registered environments (i.e. the default configurations), along with the random baseline and optimal performance. The experiments were run three times for each of the 22 registered environments, with the median represented as a blue curve and the min/max as the shaded region. Although our standard A2C agent improves upon the random baseline, optimality gaps remain in most environments (i.e., the differences between the A2C and optimal performance) highlighting challenges in solving combinatorial problems.

## 4.3 ENVIRONMENT PARALLELIZATION EXPERIMENTS

We present an initial experiment demonstrating the speed of Jumanji environments as we parallelize the step function. Figure 3a shows how the throughput of the environment step function increases

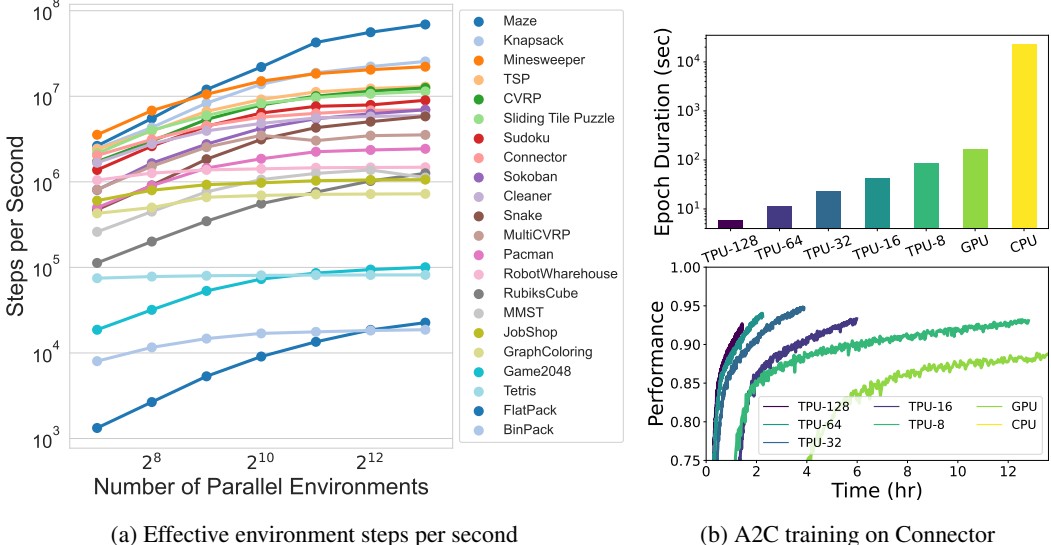

(a) Effective environment steps per second  (b) A2C training on Connector

Figure 3: Analysis of environment parallelization. (a): Scaling of the effective environment steps per second (throughput) for each registered environment as the number of parallel environments increases, on an 8-core TPU-v4. The legend is ordered by the throughput of the rightmost data point. The results on GPU and CPU are presented in Appendix C.2. (b): Training of an A2C agent on the Connector environment on a CPU, GPU (RTX 2080 super), and TPU-v4 with a number of cores varying from 8 to 128. Each training is run for 255M steps. Full training takes weeks on a CPU, which is why it is not visible on the bottom plot. Performance denotes the proportion of wires connected (an optimal policy would reach 1.0). See Appendix C.2 for further details.

with the number of environments run in parallel, on a TPUv4-8. To study parallelization on different hardware, we run a similar experiment on a GPU (V100) and a CPU in Appendix C.2. We compute the number of steps per second by averaging 50 consecutive actions each taken on 500 environments in parallel. The cost of the reset function is environment-dependent and in some cases expensive, therefore, we focus on the scaling properties of the step function and do not reset the environments. Refer to Appendix E for a discussion on parallelizing environments.

To quantify the benefits of device parallelization, we train an A2C agent on the combinatorial Connector environment varying the hardware, specifically, CPU, GPU (RTX 2080 super), and TPU with 8 to 128 cores. Figure 3b shows the approximately linear scaling of convergence speed when increasing TPU cores, demonstrating efficient parallelization across devices. For example, we can reach $92\%$ of the optimal performance on Connector-v2 in roughly 1.5h with a TPU-128, compared to 11.5h with a TPU-8.

## 5 FLEXIBILITY AND SCALABILITY IN JUMANJI

Jumanji is designed with flexibility and scalability at its core. In this section, we present initial experiments demonstrating these two key properties. In Section 5.1, we demonstrate the flexibility of Jumanji environments, by implementing specific initial state distributions via custom generators. We first discuss why flexibility is required for building robust RL agents for real-world settings, and showcase it with an initial experiment using multiple generators. In Section 5.2, we discuss how problem complexity is scalable in Jumanji and present experiments demonstrating its impacts on agent performance.

### 5.1 FLEXIBILITY IN JUMANJI

**Discussion of Jumanji Generators**   Training agents on a wide range of data distributions has been demonstrated to enhance their robustness towards real-world scenarios (Bossek et al., 2019; Cobbe et al., 2019; Taiga et al., 2023). The flexibility to define the initial state distribution provided

by generators offers two significant advantages. Firstly, it enables users to train agents on desired data distributions, by creating one or multiple custom generators to sample from. This is especially useful for combinatorial problems where there is no canonical instance distribution. Secondly, the environment dynamics are agnostic to the generator, this allows us to maintain consistent dynamics while having the flexibility to alter the initial state distribution. This facilitates experimentation and analysis on different initial state configurations, enhancing our ability to understand a given agent's behavior across various scenarios.

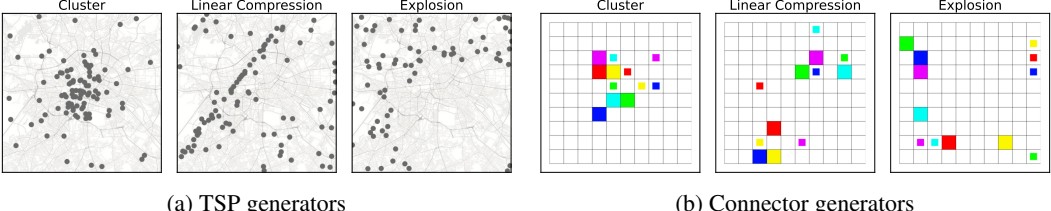

(a) TSP generators          (b) Connector generators

Figure 4: Samples from three custom generators: cluster (left), linear compression (middle), and explosion (right), for TSP and Connector environments. In TSP, gray dots represent cities. In Connector, the node pairs to be connected are depicted in the same color, and the large and small blocks indicate the starting and ending nodes, respectively.

**Example Generators**    Here, we provide illustrative examples of possible generators for two environments. Specifically, we showcase three different generators for the Traveling Salesman Problem (TSP) and Connector environments in Figure 4. In both environments, an instance consists of 2D node coordinates, with the objective being to form a brief cycle (TSP) or connect all same-type node pairs without overlap (Connector). Problem instances are created using the different generators: cluster, linear compression, and explosion. The cluster generator allocates points within a specified radius and center point, the linear compression generator randomly aligns the points along a 2D line within the space, and the explosion generator pushes the points away from a given reference point in the space.

**Experiments using Multiple Generators**    We train two A2C agents on different initial state distributions for a TSP environment comprised of 50 cities and evaluate the generalization capabilities of the resulting agents. Specifically, one agent is trained using random uniform instances, while the second agent samples from a combination of the uniform generator and the three previously introduced TSP generators. To evaluate our agents, we create two datasets from the VLSI TSP Benchmark Dataset (Rohe) [1] that contain real-world problem instances. During training, we use 102 problem instances to evaluate the agent's performance whilst at test time, we use a larger dataset of 1 020 instances.

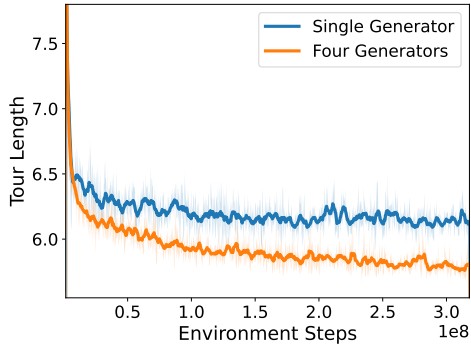

Figure 5: Learning curves of two agents training on TSP, sampling from a single uniform generator versus sampling from four generators (uniform, cluster, explosion, and compression). Lower tour length is better.

Figure 5 shows the learning curves of the two A2C agents (smaller tour length is better). At test time, on the larger set of problem instances, the agent trained on a single uniform generator achieved a mean tour length of 6.090 ($\pm 0.029$), whereas the agent trained on the four generators attained a better average tour length of 5.815 ($\pm 0.025$). These results demonstrate that the model trained on a broader data distribution, facilitated by the inclusion of multiple generators, outperforms the model trained with a single generator on an unseen, real-world test set and thus, shows better generalization capabilities. For further experimental details, please refer to Appendix C.3.

[1] https://www.math.uwaterloo.ca/tsp/vlsi/index.html

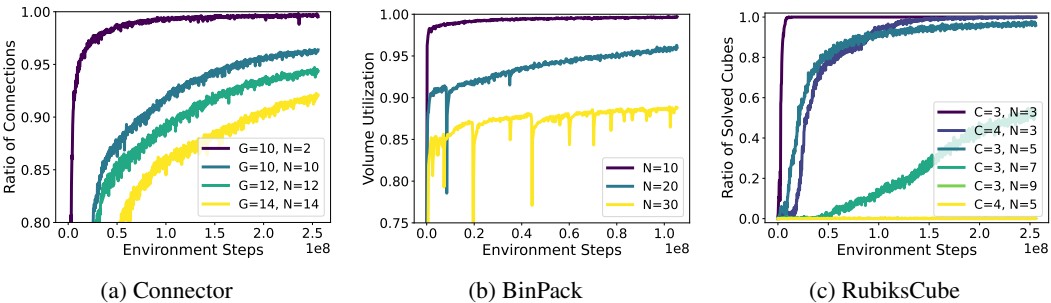

(a) Connector             (b) BinPack             (c) RubiksCube

Figure 6: The learning curves of the A2C agent on varying difficulty levels in three different Jumanji environments. In Connector, the size of the grid ($G$) and the number of node pairs to be connected ($N$) are varied. In BinPack, the number of items ($N$) is varied. In RubiksCube, the size of the cube ($C$) and the number of scrambles made from a solved Rubiks Cube ($N$) are varied.

## 5.2 SCALABILITY IN JUMANJI

Jumanji environments are scalable: each environment is equipped with one or more adjustable variables (such as the number of cities for TSP or the number of node pairs and grid size for Connector) that allow for arbitrary complexity. This flexibility is a key attribute of Jumanji environments, as it enables users to control the problem complexity and examine its impact on the agent's performance.

**Experiments varying Problem Complexity** To showcase scalability, we investigate the scaling properties of an A2C agent on three Jumanji environments: 1) Connector, where we vary the grid size and the number of node pair that need connecting; 2) BinPack, we vary the number of items to pack; 3) RubiksCube, we vary the size of the cube and the minimal number of actions required for a solution. Appendix C.4 provides additional details on these experiments.

We present learning curves of the three environments with varying difficulty settings in Figure 6. We observe a strong degradation in performance across all environments as we increase the problem complexity. For example, in RubiksCube, the hardest problem setting we experiment with leads the A2C agent to a complete failure to learn.

This experiment provides a proof of concept into scaling the complexity of Jumanji environments. It highlights how Jumanji can be used to study scaling properties of agents.

## 6 CONCLUSION

For RL research to be useful in real-world applications, challenging new benchmark environments are required. To this end, we introduce Jumanji, an open-source and diverse suite of industry-inspired RL environments that are *fast*, *flexible*, and *scalable*. Written in JAX, Jumanji environments can be parallelized and seamlessly scale with hardware (see Section 4.3). Flexibility is provided by allowing users to define custom initial-state distributions via generators (see Section 3.2 and 5.1). At the heart of Jumanji is a set of NP-hard COPs with scalable environment dynamics that facilitate industry-scale research. While Jumanji provides industry-inspired environments, capturing the full complexity of industry situations within a single benchmark remains a challenging task. Nor can a single benchmark cover the full range of possible industry situations.

Jumanji is open-source, lightweight, and easy to extend. We welcome contributions from the community. Current environments all have discrete actions, yet Jumanji supports both discrete and continuous actions. Similarly, Jumanji supports multi-agent environments but only contains single-agent implementations. Future work will expand the library to include multi-agent implementations, environments with continuous actions, and more environments representative of real-world problems, such as in the life sciences, agriculture, logistics, and beyond. By providing a diverse suite of tasks, Jumanji aims to inspire future research toward RL agents that can learn to solve a wide range of important problems.

ACKNOWLEDGMENTS

Research supported with Cloud TPUs from Google's TPU Research Cloud (TRC). We would like to thank the many open-source contributors who have contributed to the library, including Ugo Okoroafor, Randy Brown, Ole Jorgensen, Danila Kurganov, and Marta Wolinska. We also thank Thomas Barrett, Matthew Morris, Cyprien Courtot, Edan Toledo, Paul Caron, and Ian Davies for their help in the initial design of the library. Laurence Illing Midgley acknowledges support from José Miguel Hernández-Lobato's Turing AI Fellowship under grant EP/V023756/1.

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

# Appendix

## Table of Contents

## A   THE JUMANJI LIBRARY

The Jumanji library contains 22 diverse RL environments designed to be fast, flexible, and scalable. These environments are organized into three categories: logic, packing, and routing, and Table 1 shows the environments in each category. The following subsections provide a more detailed description of each environment and are sorted by the different environment categories.

Table 1: Jumanji Environments.

| Logic | Packing | Routing |
|---|---|---|
| Game2048 | BinPack | Cleaner |
| GraphColoring | FlatPack | Connector |
| Minesweeper | JobShop | CVRP |
| RubiksCube | Knapsack | Maze |
| SlidingTilePuzzle | Tetris | MMST |
| Sudoku | | MultiCVRP |
| | | PacMan |
| | | RobotWarehouse |
| | | Snake |
| | | Sokoban |
| | | TSP |

### A.1   LOGIC ENVIRONMENTS

#### GAME2048

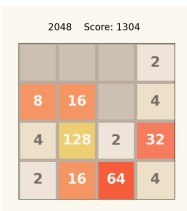

| Component | Description |
|---|---|
| Objective | Reach a high-valued tile, aiming to surpass 2048. |
| Observation | Board, action mask, and step count. |
| Action | Up (0), right (1), down (2), or left (3). |
| Reset | 4x4 grid with a single cell being either 2 or 4. |
| Reward | Sum of merged cells upon taking an action. |
| Versions | `Game2048-v1` |

#### GRAPHCOLORING

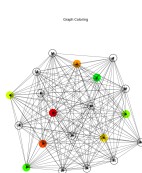

| Component | Description |
|---|---|
| Objective | Color graph vertices without adjacent vertices sharing the same color. |
| Observation | Graph, colors of the vertices, action mask, and current node. |
| Action | Integer to represent a unique color. |
| Reset | Graph with 20 nodes and a 0.8 edge probability. |
| Reward | Negative of the number of unique colors used for all vertices. |
| Versions | `GraphColoring-v0` |

#### MINESWEEPER

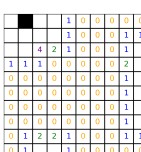

| Component | Description |
|---|---|
| Objective | Clear the board without detonating any mines. |
| Observation | Board, action mask, number of mines, and step count. |
| Action | Coordinates of the square to explore. |
| Reset | Uniformly samples 10 mines in a 10x10 grid. |
| Reward | 1 reward for a square without a mine, and 0 otherwise. |
| Versions | `Minesweeper-v0` |

## RUBIKSCUBE

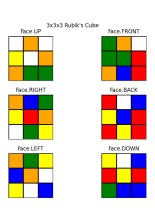

| Component | Description |
|---|---|
| Objective | Match all stickers on each face to a single color. |
| Observation | View of the current cube state and the step count. |
| Action | Tuple representing: face, depth, and direction of the turn. |
| Reset | Applies a number of scrambles to a 3x3 solved cube. |
| Reward | Reward of 1 for solving the cube and otherwise 0. |
| Optimal Return | Solved ratio equal to 1.0. |
| Versions | `RubiksCube-v0, RubiksCube-partly-scrambled-v0` |

## SLIDINGTILEPUZZLE

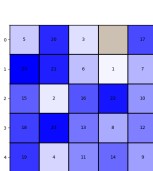

| Component | Description |
|---|---|
| Objective | Move tiles to the adjacent empty slot until the puzzle is sorted. |
| Observation | Puzzle, position of the empty tile, and step count. |
| Action | Up (0), right (1), down (2), or left (3). |
| Reset | Applies a number of random swaps to a 5x5 solved puzzle. |
| Reward | Reward of 1 for newly correct tiles and -1 for newly wrong ones. |
| Optimal Return | `prop_correctly_placed` ratio equal to 1.0. |
| Versions | `SlidingTilePuzzle-v0` |

## SUDOKU

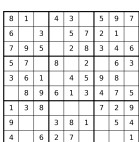

| Component | Description |
|---|---|
| Objective | Fill $N \times N$ grid with digits 1 to $N$ in each column, row, and subgrid. |
| Observation | Board, and the action mask. |
| Action | Tuple representing the square coordinates and the digit. |
| Reset | Uniformly samples a puzzle database. |
| Reward | Reward is 1 if the board is correctly solved, and 0 otherwise. |
| Optimal Return | Solved ratio equal to 1.0. |
| Versions | `Sudoku-v0, Sudoku-very-easy-v0` |

## A.2  PACKING ENVIRONMENTS

## BINPACK

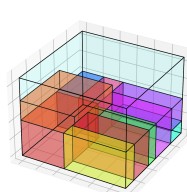

| Component | Description |
|---|---|
| Objective | Pack boxes into a container to minimize empty space. |
| Observation | Available space, set of unpacked items, and action mask. |
| Action | Tuple representing the EMS (space) ID and the item ID. |
| Reset | Randomly splits a container into different items. |
| Reward | Volume utilization of the container (between 0.0 and 1.0). |
| Optimal Return | Volume utilization equal to 1.0. |
| Versions | `BinPack-v2` |

## FLATPACK

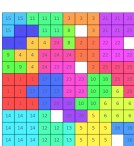

| Component | Description |
| --- | --- |
| Objective | 2D version of BinPack, place all the available blocks on a grid. |
| Observation | Current grid, available blocks. |
| Action | Block to place, rotation to make, coordinates on the grid. |
| Reset | Random instances guaranteed to be solvable. |
| Reward | Dense, fraction of the grid covered by the block. |
| Versions | `FlatPack-v0` |

## JOBSHOP

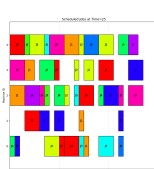

| Component | Description |
| --- | --- |
| Objective | Minimize the time needed to process all the jobs. |
| Observation | Machines, operation details for each job, and action mask. |
| Action | Array containing a job ID for each machine. |
| Reset | Instances with a number of jobs, machines, operations, and max. duration. |
| Reward | Reward of -1 at each time step if not terminating. |
| Versions | `JobShop-v0` |

## KNAPSACK (KOOL ET AL., 2018)

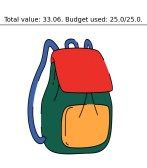

| Component | Description |
| --- | --- |
| Objective | Maximize value by packing items within weight constraint. |
| Observation | Weights, value, and packed status of the items. |
| Action | Integer to represent the next item to pack. |
| Reset | Uniformly samples item weights & values from a unit square. |
| Reward | Sum of the values of the items packed in the bag. |
| Versions | `Knapsack-v1` |

## TETRIS

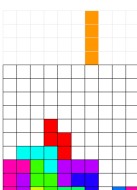

| Component | Description |
| --- | --- |
| Objective | Score maximum points by clearing lines in Tetris. |
| Observation | Grid, Tetromino, and action mask. |
| Action | Tuple denoting the x-position and rotation of the block. |
| Reset | Randomly samples Tetrominos from a predefined list. |
| Reward | Proportional to the number of lines cleared. |
| Versions | `Tetris-v0` |

## A.3 ROUTING ENVIRONMENTS

### CLEANER

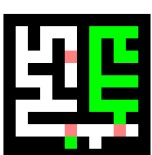

| Component | Description |
| --- | --- |
| Objective | Clean as many tiles as possible in a given time budget. |
| Observation | Grid, agent location, action mask, and step count. |
| Action | Array denoting an action (left, right, up, down) per agent. |
| Reset | Randomly generates the structure of the grid. |
| Reward | The number of tiles cleaned at each timestep. |
| Versions | `Cleaner-v0` |

### CONNECTOR

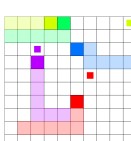

| Component | Description |
| --- | --- |
| Objective | Connect each start and end position and minimize steps. |
| Observation | Grid, action mask, and step count. |
| Action | Array with an action (left, right, up, down, no-op) per agent. |
| Reset | Uniform randomly places start and end positions on the grid. |
| Reward | 1 for connecting agents, $-0.03$ for non-connected agents. |
| Optimal Return | Ratio of connections equal to 1.0. |
| Versions | `Connector-v2` |

### CVRP (KOOL ET AL., 2018)

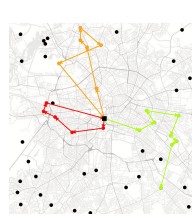

| Component | Description |
| --- | --- |
| Objective | Find shortest route visiting each city once and covering the demands. |
| Observation | Coordinates, demands, current position, and vehicle capacity. |
| Action | Integer representing the next city or depot to visit. |
| Reset | Uniformly samples coordinates and demands. |
| Reward | Negative tour length of the route. |
| Versions | `CVRP-v1` |

### MAZE

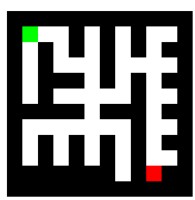

| Component | Description |
| --- | --- |
| Objective | Reach the single target cell. |
| Observation | Maze, agent and target position, action mask, and step count. |
| Action | Up (0), right (1), down (2), or left (3). |
| Reset | Randomly generates the structure of the maze. |
| Reward | 1 for reaching target, 0 otherwise. |
| Versions | `Maze-v0` |

## MMST

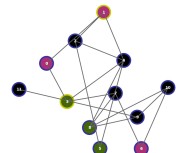

| Component | Description |
| --- | --- |
| Objective | Connect all same-type nodes without using the same utility nodes. |
| Observation | Node types, adjacency matrix, action mask, and current agent position. |
| Action | Integer array to represent the next node per agent. |
| Reset | Randomly splits the graph into multiple sub-graphs. |
| Reward | 10 for valid connection, $-1$ for no connection, $-1$ for invalid action. |
| Versions | `MMST-v0` |

## MULTICVRP

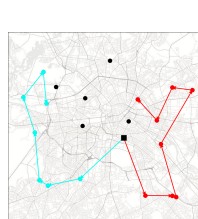

| Component | Description |
| --- | --- |
| Objective | Route multiple agents to satisfy the demands of all cities. |
| Observation | Coordinates, demands, time windows, penalties agent locations, local times, and capacities. |
| Action | Integer array to represent the next city for each agent. |
| Reset | Uniformly samples coordinates and demands. |
| Reward | The negative tour length of all agents combined. |
| Versions | `MultiCVRP-v0` |

## PACMAN

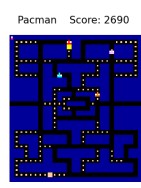

| Component | Description |
| --- | --- |
| Objective | Collect all the pellets and avoid the ghosts. |
| Observation | Grid, agent location, ghost location, pellet location, power-pellet locations, and time left for scatter state. |
| Action | Up (0), right (1), down (2), left (3), or no-op (4). |
| Reset | Deterministic generator to start state of the game. |
| Reward | +10 for each pellet, +20 for a power pellet, +200 for a ghost. |
| Versions | `PacMan-v0` |

## ROBOTWAREHOUSE (PAPOUDAKIS ET AL., 2021)

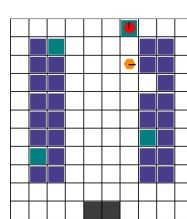

| Component | Description |
| --- | --- |
| Objective | Deliver as many requested shelves in a given time budget. |
| Observation | View of other agents & shelves, action mask, and step count. |
| Action | No-op (0), forward (1), left (2), right (3), or toggle load (4) per agent. |
| Reset | Randomly places agents on the grid and uniformly selects shelves. |
| Reward | Number of shelves delivered during the timestep. |
| Versions | `RobotWarehouse-v0` |

SNAKE

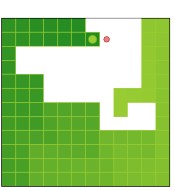

| Component | Description |
|---|---|
| Objective | Collect as many fruits without colliding with its own body. |
| Observation | Grid, action mask, and step count. |
| Action | Up (0), right (1), down (2), or left (3). |
| Reset | Randomly places snake's head position and fruit on the grid. |
| Reward | Reward is +1 upon collection of a fruit and 0 otherwise. |
| Optimal Return | Equal to 143. |
| Versions | `Snake-v1` |

SOKOBAN (WEBER ET AL., 2018; GUEZ ET AL., 2019; SCHRADER, 2018)

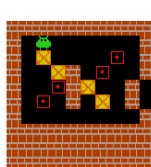

| Component | Description |
|---|---|
| Objective | Box-pushing environment where the goal is to place all boxes on their targets. |
| Observation | Grid, and step count. |
| Action | Up (0), right (1), down (2), or left (3). |
| Reset | Sample from a dataset (Guez et al., 2019). |
| Reward | -0.1 for each step, +1/-1 for each correct/incorrect box, +10 if success. |
| Versions | `Sokoban-v0` |

TSP (KOOL ET AL., 2018)

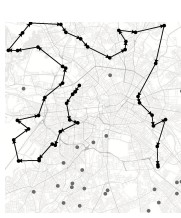

| Component | Description |
|---|---|
| Objective | Find shortest route, visit each city once, and return to starting city. |
| Observation | Coordinates, current position, trajectory, and action mask. |
| Action | Integer representing the next city to visit. |
| Reset | Uniformly samples coordinates from a unit square. |
| Reward | Negative tour length. |
| Versions | `TSP-v1` |

## A.4 EXTENDING THE LIBRARY

Below is a template of how to extend Jumanji's `Environment` interface to create a new environment:

```python
from typing import Tuple, NamedTuple

import chex
from chex import dataclass
import jax

from jumanji import specs
from jumanji.env import Environment
from jumanji.types import TimeStep, restart, termination, transition

@dataclass
class State:
    ...
    key: chex.PRNGKey

```

```python
18  class Observation(NamedTuple):
19      ...
20
21
22  class MyEnv(Environment[State]):
23      def __init__(self):
24          ...
25
26      def reset(
27          self, key: chex.PRNGKey
28      ) -> Tuple[State, TimeStep[Observation]]:
29          observation = Observation(...)
30          state = State(key=key, ...)
31          timestep = restart(observation)
32          return state, timestep
33
34      def step(
35          self, state: State, action: chex.Array
36      ) -> Tuple[State, TimeStep[Observation]]:
37          next_state = State(key=state.key, ...)
38          done = ...
39          reward = ...
40          next_observation = Observation(...)
41
42          next_timestep = jax.lax.cond(
43              done,
44              termination,
45              transition,
46              reward,
47              next_observation,
48          )
49
50          return next_state, next_timestep
51
52      def observation_spec(self) -> specs.Spec[Observation]:
53          obs_spec = ...
54          return obs_spec
55
56      def action_spec(self) -> specs.Spec:
57          action_spec = ...
58          return action_spec
```

After implementing a new environment, one may optionally add it to the registry using `jumanji.register`. This allows users to then instantiate the newly implemented environment with `jumanji.make`.

```python
1  from jumanji.registration import register
2
3  register(id="MyEnv-v0", entry_point="...:MyEnv", kwargs=...)
4
5  env = jumanji.make("MyEnv-v0")
```

**Wrappers** Jumanji has several wrappers that one can use to convert a Jumanji environment to the API of one's choice. For instance, one can use `JumanjiToDMEnvWrapper` to make a `dm_env` environment, or `JumanjiToGymWrapper` to convert it to the `gym` API.

## A.5 Multi-agent RL with Jumanji

Jumanji includes several environments whose action space is multi-dimensional, such as Robot Warehouse, an existing multi-agent environment (Papoudakis et al., 2021), as well as Connector, MultiCVRP, MMST and Cleaner. These can be seen as homogeneous multi-agent environments where each agent is responsible for a scalar action. For instance, Connector has $N$ heads that need to connect to their target. One can see it as a single-agent environment with an action of shape $(N,)$ (one value per head), or a multi-agent environment where each head is its own agent and outputs a scalar action. This is called a homogeneous multi-agent environment because each agent has the same observation and action shapes. However, Jumanji does lack a true multi-agent training algorithm, as the intention is only to provide reasonable benchmarks. Instead, when training an agent on a multi-agent environment, Jumanji trains in the style of centralized training with centralized execution (Lowe et al., 2017) and treats the environment as a single-agent one.

## B Networks

We provide an implementation of an advantage actor-critic (A2C) agent running on each environment in Jumanji. To do so, the algorithm is made agnostic to the environment and only assumes an environment-specific actor-critic network that, given an observation, outputs a policy over actions and a value of the current state. This means these networks can be used for any actor-critic algorithm that uses state-value functions (e.g. A2C, PPO, TRPO, etc).

Each environment comes with its own set of symmetries such as invariance to index permutations and is represented using a specific geometry, e.g. grids/images, sets, etc. Some symmetry groups are very large, e.g. TSP is invariant to permutation of the node indices; such permutations form a group of size $n!$ where $n$ is the number of cities. It is inconceivable to hope to statistically learn a good policy for each of these permutations. Instead, we include geometrical biases in the network architectures to get these symmetries for free. For instance, we make the TSP policy network equivariant to city permutations and the critic network invariant to those permutations.

We open-source actor-critic networks for each environment in `jumanji/training/networks` along with their configs in `jumanji/training/configs/env`. We list below some of the symmetries existing in each environment and describe what network is implemented as a consequence.

**BinPack** The observation is composed of two sets: the spaces (EMS) and the items to pack in the container. Therefore, we use an independent self-attention layer for each set and then use cross-attention between each set based on whether an item fits in the corresponding space EMS). Then, embeddings of both sets are joined using an outer product to ensure permutation equivariance (or invariance for the critic) within both items and spaces.

**Cleaner** The grid is first copied $n$ times where $n$ is the number of agents. Each agent sees a version of the grid where it is colored differently from the other agents. Then, a CNN is vmap-ed over the different grids and outputs $n$ feature vectors that are all passed through an MLP to output logits for each agent. The network is equivariant to permutations of agent locations.

**Connector** The network is similar to Cleaner's in processing the agent grids independently via a CNN first. Then, as opposed to Cleaner, the $n$ feature vectors are passed to a transformer so that each agent can attend to one another in a permutation-equivariant way.

**CVRP** The network is adapted from Kool et al. (2018) to have a transformer encoder part that encodes all non-visited nodes and a decoder that includes the current position to determine the next action. The important symmetry to respect here is equivariance to permutations of nodes.

**FlatPack** The observation is permutation invariant with respect to the order of blocks. Therefore, we use a sequence model (transformer) to process all the blocks and obtain a permutation-equivariant policy. The grid is processed using a small U-net.

**Game2048** The observation being an image, we use a CNN with valid padding to prevent modeling the board edge the same way as empty cells.

**GraphColoring**    The observation contains nodes and colors. They are represented as two sequences and the graph-coloring problem is invariant to permutations of both nodes and colors. Consequently, the provided network uses transformer blocks alternating between self-attention on each sequence and cross-attention between these sequences. Alternatively, a GNN could be implemented instead given the graph structure of the problem.

**JobShop**    The observation contains two sequences with respect to which the problem is invariant to permutations of indices: jobs and machines. Each job is itself a sequence of operations that have to run on a specific machine. The problem is also invariant to renaming these indices by permuting machines. To leverage these symmetries, we build a network that does cross-attention between the operations and the machines they have to run on for each job (parallel across jobs). Then, these operations sequences are reduced (averaged) to provide a single job embedding for each job. Cross-attention between jobs and machines leads to the action distribution (resp. value estimation) in a permutation-equivariant (resp. invariant) way.

**Knapsack**    The problem is invariant to permutations of the items. The implemented network is a transformer that is also adapted from Kool et al. (2018) and uses self-attention on the remaining items.

**Maze**    The observation is a grid/image, so we implement a CNN to process the grid before passing it through an MLP to obtain the action logits or value estimate.

**Minesweeper**    Same as the Maze, Minesweeper has a grid observation processed by a CNN.

**MMST**    The observation contains information about two sequences: the different agents and the nodes on the graph. The problem is invariant to permutations of agent IDs and node indices. Hence, we implement a transformer network that alternates between self-attention layers on each sequence and cross-attention between the agents and their nodes.

**MultiCVRP**    The vehicles and the customers are first encoded. Then a series of self-attention and cross-attention is used on both sequences.

**PacMan**    We use a CNN to process the grid image. We then concatenate the grid embedding with diverse observation features like the agent's position and the ghosts'. A final MLP head projects these embeddings to a value (critic) or logits (actor).

**RobotWarehouse**    The observation contains a feature vector for each agent. They are processed as a sequence by a transformer to be equivariant with respect to permutations of agents.

**RubiksCube**    The cube is just flattened and then passed through an MLP. The network would probably benefit from a more symmetry-preserving architecture for this environment. Yet, it is not obvious how to do so.

**SlidingTilePuzzle**    We use a CNN to process the grid and then an MLP to project to value and logits.

**Snake**    The observation is an image with 5 feature maps. Therefore, we process it with a CNN before using an MLP to output logits or a value estimate.

**Sokoban**    A CNN processes the grid, and a final MLP head projects the grid embedding and the step count to a value or logits.

**Sudoku**    The environment has many symmetries, including permutations of digits (e.g. 3 and 6 are swapped), permutations of columns within a 3-column group, etc. We design a network that is equivariant to the first symmetry. We flatten the grid and use this as a feature vector for each digit. We then do self-attention in the digit sequence to respect the permutation equivariance and then transpose back to the grid dimension.

**Tetris** The observation is composed of the grid and the tetromino. We process the former with a CNN to which we concatenate the flattened tetromino processed by an MLP. Last, an MLP head outputs action logits or a value estimate.

**TSP** The network is almost equivalent to CVRP but does not include a depot node. The symmetry that is respected is the permutation to node indices.

## C EXPERIMENTS

### C.1 ACTOR-CRITIC BASELINE

| Name | Version | Wall-clock time |
|------|---------|-----------------|
| BinPack | `"BinPack-v2"` | 30h |
| Cleaner | `"Cleaner-v0"` | 11h |
| Connector | `"Connector-v2"` | 35h |
| CVRP | `"CVRP-v1"` | 3h |
| FlatPack | `"FlatPack-v0"` | 48h |
| Game2048 | `"Game2048-v1"` | 3h |
| GraphColoring | `"GraphColoring-v0"` | 7.5h |
| JobShop | `"JobShop-v0"` | 20 min |
| Knapsack | `"Knapsack-v1"` | 10 min |
| Maze | `"Maze-v0"` | 30 min |
| Minesweeper | `"Minesweeper-v0"` | 3h |
| MMST | `"MMST-v0"` | 6h |
| MultiCVRP | `"MultiCVRP-v0"` | 1.5h |
| PacMan | `"PacMan-v0"` | 4.5h |
| RobotWharehouse | `"RobotWarehouse-v0"` | 6.5h |
| RubiksCube | `"RubiksCube-partly-scrambled-v0"` | 3h |
| SlidingTilePuzzle | `"SlidingTilePuzzle-v0"` | 40 min |
| Snake | `"Snake-v1"` | 1h |
| Sokoban | `"Sokoban-v0"` | 3.5h |
| Sudoku | `"Sudoku-very-easy-v0"` | 3.5h |
| Tetris | `"Tetris-v0"` | 3.5h |
| TSP | `"TSP-v1"` | 2.5h |

Table 2: Correspondance between the name reported in figure 3a legend and the environment version.

We train an actor-critic agent on each environment on 3 different seeds in figure 2. For this, we use the registered versions displayed in table 2, the open-sourced networks described in section B, and the configs available on GitHub as well.

We use the `train.py` script from `https://github.com/instadeepai/jumanji/blob/main/jumanji/training/train.py` that alternates between evaluating the agent and training it. Each training epoch consists of a number of `num_learner_steps_per_epoch` of collecting `n_steps` on `total_batch_size` environments in parallel. If multiple devices are available, the batch of environments is split between the accelerators, on which trajectories are collected directly using a local copy of the model. After collecting trajectories, we compute an A2C loss and update the parameters.

The A2C loss is a weighted mixing of three terms: a policy gradient term $-A(\tau) \log \pi_\theta(\tau)$, a critic term $A(\tau)^2$ and an entropy bonus $-\mathcal{H}(\pi_\theta(\tau))$

We release checkpoints for all the agents on Hugging Face Hub.

### C.2 ENVIRONMENT PARALLELIZATION EXPERIMENTS

This section provides more details on the parallelization experiments described in section 4.3. The first experiment aims to demonstrate how the environment dynamics can be parallelized to increase the steps throughput, and the second experiment shows how a full training pipeline can benefit from parallelization by reducing the time to optimality.

**Environment Parallelization**   This experiment shows how the raw environments' speed increases with the number of environments that are run in parallel. The number of environments, gradually increasing from 128 to 8192, is equally divided among 8 TPUv4 cores. To evaluate the raw speed of the environment dynamics, only the duration of the actual step function has been considered. Starting from a generated initial state, the same action is applied 50 times in all the parallel environments. This procedure is run 500 times to form an epoch. The total number of steps ran in an epoch (50 multiplied by 500 multiplied by the number of parallel environments) is divided by the epoch duration to obtain the average number of steps per second. The throughput which is reported in figure 3a is the average over the second epoch run, the first epoch being longer as biased by the JIT compilation time which is only run once and thus is not representative of the final environment dynamics performance. The same experiment is executed on GPU (V100) and CPU as well, and the results are shown in Figure 7. Table 2 contains the version of each environment in which speed was measured in this experiment.

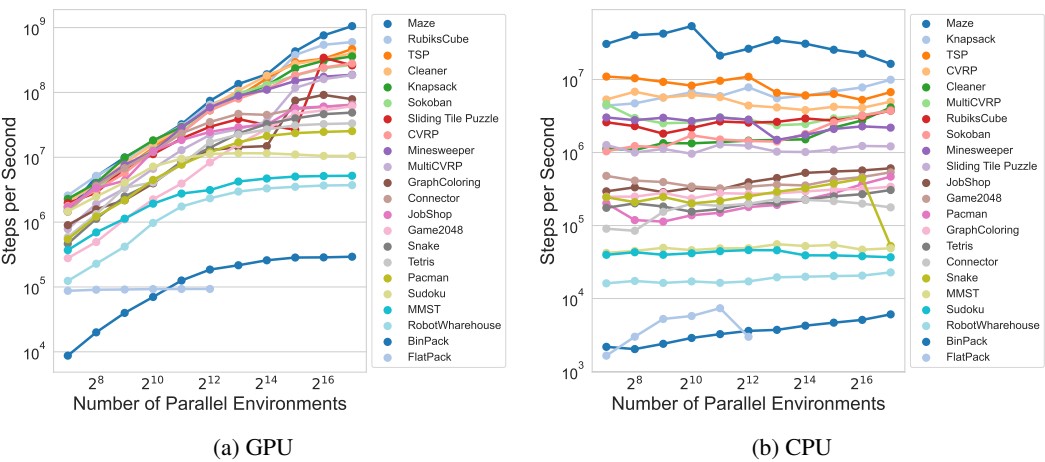

(a) GPU                                                                (b) CPU

Figure 7: Scaling of the effective environment steps per second (throughput) for each registered environment as the number of parallel environments increases on (a) GPU (Tesla V100) and (b) CPU (2 cores).

**Hardware scaling of A2C training**   This experiment shows the impact of increasing the training resources of the Jumanji A2C agent on the Connector-v2 environment. The training was done on different hardware: 1 CPU with 8 cores, 1 GPU (RTX 2080 super) and a TPU-v4 with a varying number of cores, i.e. 8, 16, 32, 64, and 128 cores. The training is run for 1000 epochs of 100 learning steps in which 256 trajectories of length 10 are sampled. The sampling of the trajectories is split across the available devices, but the number of environment steps sampled per epoch is the same for all the training settings. The A2C agent is run without normalizing advantages, with a discount factor of 1, a bootstrapping factor of 0.95, and a learning rate of $2 \times 10^{-4}$. We demonstrate almost linear scaling in hardware in table 3 by plotting training convergence speed as a function of the number of TPU cores.

| TPU cores | 8 | 16 | 32 | 64 | 128 |
|---|---|---|---|---|---|
| Convergence time (h) | 12.8 | 4.2 | 3.2 | 2.0 | 1.4 |

Table 3: Time to reach 92% performance as a function of the TPU cores. This table completes the experiment presented in figure 3b.

## C.3   FLEXIBILITY EXPERIMENTS

This section provides additional details on the experiments conducted using multiple generators for the TSP problem (5.1).

**Network**    The networks of the two A2C agents in this experiment are identical, and additionally, it is the same network used in the baseline experiment for the TSP environment (section 4.2). Full details of the network used in this experiment can be found in Appendix B.

**Training Procedure**    The A2C agent trained on the single uniform generator was trained in an identical manner as the A2C agent for the TSP baselines experiment, this includes the same training hyper-parameters (e.g, the sequence length, batch size, and so forth). The only notable difference between the training process of the two agents is that the baseline TSP agent is trained on 50 cities whereas the agent for this experiment is trained on 20 cities. The A2C agent trained on the combination of the uniform generator and the three custom generators (cluster, linear compression, and explosion). The custom generators were implemented by inheriting the abstract generator class from their environment and then modifying the `call` method to return instances with the desired initial state distribution. This A2C agent is trained in a similar manner with the following difference. The batch of data used to update the single generator agent contains the agent's trajectories on only uniform instances, whereas, the four generators agent's batch of data consists of its trajectories on uniform, cluster, linear compression, and explosion instances. In the former, the agent interacts solely in the environment with the uniform generator, whereas in the latter, the agent sequentially interacts with four environments each with a specific (uniform or custom) generator. Lastly, both of the agents were trained for 300 million environment steps.

**Evaluation Dataset**    The dataset was created using the VLSI TSP Benchmark Dataset. There are 102 TSP instances in the VLSI dataset with instance size (i.e., number of cities) ranging from 131 to 744710. Since we conduct this experiment with a TSP environment with 50 cities, we randomly sample 50 cities from each of the 102 instances to obtain an unseen, real-world validation dataset of 102 instances each with 50 cities. This dataset is used to evaluate the A2C agents during training. Additionally, we create a larger dataset with 1020 instances by randomly sampling 50 cities from each of the 102 instances 10 times. This larger dataset is used to create a lower-variance estimate of the performance of both agents at test time.

**Results**    Section 5.1 presents the results of the two A2C agents on the unseen, real-world instances, both during training and at test time, and it can be seen that the agent trained on a broader initial state distribution (i.e., four generators agent) outperformed the agent trained solely on uniform instances. To further analyze the two agents, they were also evaluated on a random set of uniform instances during training, and Figure 8 depicts the learning curves of the agents. The aim of this evaluation was to determine the impact of training on a broader data distribution when evaluating on in-training distribution instances. Even though it does appear that the learning curve of the four generators agent is lower/better than the single generator agent's curve, this difference in performance is not significant. Therefore, these results suggest that training on a wider data distribution does not lead to any significant improvement or degradation of performance when evaluating on instances seen during training.

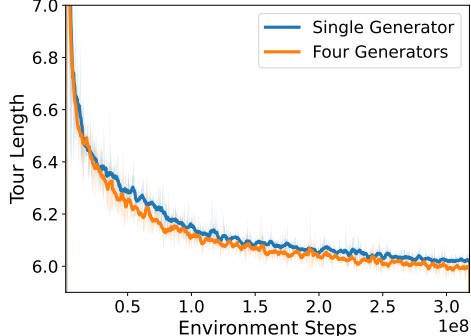

Figure 8: Learning curves of two agents training on TSP and evaluated on instances sampled from a uniform generator. One agent (blue) samples from a single uniform generator versus the other agent (orange) samples from four generators (uniform, cluster, explosion, and compression). Lower tour length is better.

### C.4    SCALABILITY EXPERIMENTS

This section provides additional details on the experiments conducted with varying degrees of problem complexity for the Connector, BinPack, and RubiksCube environments (5.2).

**Networks**    The agents in the scalability experiments are actor-critic networks which are fully defined in Appendix B. Therefore, the agents for each environment have identical networks with the sole difference being the complexity level of the environment they are trained on.

**Training** The training process is identical to the baselines experiments, we use the same training hyper-parameters and use the `jumanji/training/train.py` script in the same manner as described in Appendix C.1. The only difference is that for the purpose of the scalability experiments, we instantiate the Connector, BinPack and RubiksCube environments with different configurations (whereas for the baselines experiment, the default environment parameters configuration is used). The code snippets below show the instantiation of the different environments for the three problems.

### Connector setup

```python
from jumanji.environments import Connector
from jumanji.environments.routing.connector.generator import RandomWalkGenerator

env_1 = Connector(generator=RandomWalkGenerator(grid_size=10, num_agents=2))
env_2 = Connector(generator=RandomWalkGenerator(grid_size=10, num_agents=10))
env_3 = Connector(generator=RandomWalkGenerator(grid_size=12, num_agents=12))
env_4 = Connector(generator=RandomWalkGenerator(grid_size=14, num_agents=14))
```

### BinPack setup

```python
from jumanji.environments import BinPack
from jumanji.environments.packing.bin_pack.generator import RandomGenerator

env_1 = BinPack(
    generator=RandomGenerator(
        max_num_items=10, max_num_ems=15, split_num_same_items=2,
    ),
    obs_num_ems=15,
)
env_2 = BinPack(
    generator=RandomGenerator(
        max_num_items=20, max_num_ems=40, split_num_same_items=2,
    ),
    obs_num_ems=40,
)
env_3 = BinPack(
    generator=RandomGenerator(
        max_num_items=30, max_num_ems=60, split_num_same_items=2,
    ),
    obs_num_ems=50,
)
```

### RubiksCube setup

```python
from jumanji.environments import RubiksCube
from jumanji.environments.logic.rubiks_cube.generator import ScramblingGenerator

env_1 = RubiksCube(
    generator=ScramblingGenerator(cube_size=3, num_scrambles_on_reset=3),
)
env_1 = RubiksCube(
    generator=ScramblingGenerator(cube_size=4, num_scrambles_on_reset=3),
)
env_1 = RubiksCube(
    generator=ScramblingGenerator(cube_size=3, num_scrambles_on_reset=5),
)
env_1 = RubiksCube(
    generator=ScramblingGenerator(cube_size=3, num_scrambles_on_reset=7),
)
env_1 = RubiksCube(
```

```
17        generator=ScramblingGenerator(cube_size=3, num_scrambles_on_reset=9),
18    )
19    env_1 = RubiksCube(
20        generator=ScramblingGenerator(cube_size=4, num_scrambles_on_reset=5),
21    )
```

Table 4: Results of the scalability experiments.

| Environment | Configuration | Complexity Level | Final Performance |
|---|---|---|---|
| Connector | G = 10, N = 2 | Easy | 0.983 |
| | G = 10, N = 10 | Medium | 0.885 |
| | G = 12, N = 12 | Medium | 0.855 |
| | G = 14, N = 14 | Difficult | 0.785 |
| BinPack | N = 10 | Easy | 0.993 |
| | N = 20 | Medium | 0.934 |
| | N = 30 | Difficult | 0.870 |
| RubiksCube | C = 3, N = 3 | Easy | 0.987 |
| | C = 4, N = 3 | Medium | 0.829 |
| | C = 3, N = 5 | Medium | 0.838 |
| | C = 3, N = 7 | Difficult | 0.213 |
| | C = 3, N = 9 | Difficult | 0.000 |
| | C = 4, N = 5 | Difficult | 0.000 |

**Results** We further describe the results obtained from the scalability experiments in Table 4. This table shows the different configurations for each environment along with its qualitative definition of the complexity level and shows the final performance obtained by the A2C agent for each environment and configuration. It can be seen that with the increasing complexity of the environment, the agent performance worsens.

## D ROLL OUT THE ENVIRONMENT

### D.1 ANIMATE AN EPISODE

Below is a code example of how to take random actions in the BinPack environment and animate an episode. This code can run in a notebook or e.g. on Google Colab. Please see the `load_checkpoints.ipynb` notebook on `https://github.com/instadeepai/jumanji/blob/main/examples/load_checkpoints.ipynb` to load pre-trained agents or roll out a random policy.

```
1    %pip install --quiet jumanji[train]
2
3    %matplotlib notebook
4    import jax
5    import jumanji
6    from jumanji.training import networks
7
8    env = jumanji.make("BinPack-v2")
9    reset_fn = jax.jit(env.reset)
10   step_fn = jax.jit(env.step)
11   random_policy = networks.make_random_policy_bin_pack(env.unwrapped)
12
13   @jax.jit
14   def select_random_action(observation, key):
15       """Call `random_policy` which expects a batch of observations."""
16       batched_observation = jax.tree_util.tree_map(
```

```
17          lambda x: x[None], observation,
18      )
19      return random_policy(batched_observation, key).squeeze(axis=0)
20
21  key = jax.random.PRNGKey(0)
22  state, timestep = reset_fn(key)
23  states = [state]
24
25  # Loop until the episode is done.
26  while not timestep.last():
27      # Select an action.
28      action_key, key = jax.random.split(key)
29      action = select_random_action(timestep.observation, action_key)
30      # Step in the environment.
31      state, timestep = step_fn(state, action)
32      states.append(state)
33
34  env.animate(states, interval=100)
```

## D.2 PLACEMENT OF THE ENVIRONMENT ON DEVICE

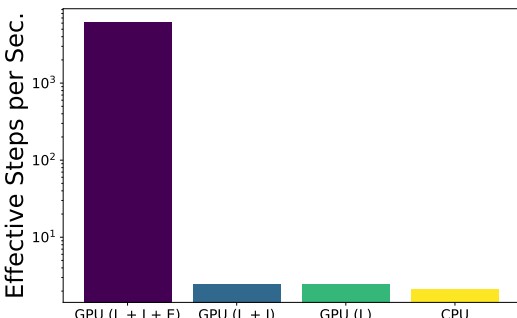

Figure 9: Throughput ablation of the full Anakin GPU setup. The experiment uses the Connector environment and is run on a V100 GPU with a batch size of 2048. The pipeline alternates between phases of acting (policy inference and environment step) and learning (back-propagation). **CPU**: The whole Jumanji pipeline is run on the CPU. **GPU (L)**: The inference and environment step remain on the CPU but the data is sent to the GPU for the learning step (L). **GPU (L + I)**: Only the environment step is done on CPU. The inference (I) and the learning step are done on the GPU. **GPU (L + I + E)**: The Anakin implementation. The whole training pipeline is performed on GPU, now including the environment step (E).

Implementing environments in Jax achieves its fullest potential when using the Anakin (Hessel et al., 2021) architecture for fully optimizing device accelerators. The pipeline consists of a synchronous execution of actor inference (action selection), environment step, and learner step (back-propagation) all on the device (e.g. GPU, TPU). High throughput is achieved by removing any host-device communication (e.g. CPU to GPU) during the process. We run an ablation study in figure 9 to study the speed-up that comes with implementing the environment in Jax and running it on the device. We use the Connector environment with a batch size of 2048. We observe that most of the speed gain arises from running the environment on GPU and not transferring data between the host and the device.

# E  DISCUSSION ON PARALLELIZATION

## E.1  PARALLELIZING THE DYNAMICS

Although the `Environment` framework within Jumanji is agnostic to using discrete or continuous actions, all of the implemented environments use a discrete action space. Having discrete actions often leads to the impossibility of fully parallelizing the dynamics, i.e. the environment step function. For instance, in Sudoku, there are 18 possible actions (6 faces and 3 different rotations). Each action leads to doing a different operation on the cube (3D array). When the step function is vmap-ed, the discrete choice of selecting the rotation to perform as a function of the action is not parallelizable within the SIMD (single instruction, multiple data) paradigm. Therefore, vmap-ing the dynamics leads to transforming the conditional branching to a select XLA statement. This means each of the 18 actions is performed for the whole batch and then the correct rotation is selected based on the action.

When executed, the code below shows how JAX's `jax.lax.cond` is transformed into a `select` when the function is vmap-ed.

```
1  import jax
2  import jax.numpy as jnp
3
4  def f(x, bool_):
5      return jax.lax.cond(bool_, lambda a: a+100, lambda a: a, x)
6
7  args = jnp.array(0, float), jnp.array(False)
8  print(jax.xla_computation(f)(*args).as_hlo_text())
9  print("---")
10 print(
11     jax.xla_computation(jax.vmap(f))(
12         jnp.array([0], float), jnp.array([False]),
13     ).as_hlo_text()
14 )
```

Because the vmap-ed dynamics have to generate all possible actions for the whole batch, it may explain why some environments end up being slower than expected on a hardware accelerator. Yet, the use of hardware-accelerated environments really shines when training a neural network as the policy since we avoid transferring data between the CPU and the accelerator.

## E.2  PARALLELIZING AUTO-RESET DURING TRAINING

During training, we roll out a few steps on a batch of parallel environments with an automatic reset behavior. This means any of the environments that reaches a terminal state is automatically reset to an initial state (with a discount of 0).

In Jumanji, we implement an environment wrapper called `AutoResetWrapper` to do this auto-reset automatically. This wrapper first calls the environment step function, then checks if it reaches a terminal state and if so, it resets the environment. Similar to explained above, this conditional statement is not parallelizable when used with vmap. As a consequence, if the wrapper is vmap-ed (for instance by wrapping it into Jumanji's `VmapWrapper`), both branches (resetting and not resetting) will be executed on all environments across the batch, at every timestep. This may be very slow if the environment reset is a slow function, which is the case for Rubikscube where the reset function is literally 100 times as computationally heavy as the step function.

An alternative to calling the reset function at every step is to use Jumanji's `VmapAutoResetWrapper` that is equivalent to vmap-ing the auto-reset behavior but actually only vmaps the step function and then loop over the environments to reset the ones that reach a terminal state. This way, if none of the states in the batch has terminated, the wrapper will not call reset once, compared to the previous wrapper which would still call reset on the whole batch.

## F  LICENSE

Jumanji is released under an Apache License 2.0.

