# OpenReview forum: "Jumanji: a Diverse Suite of Scalable Reinforcement Learning Environments in JAX"
_ICLR.cc/2024/Conference — ICLR 2024 poster_

### Official Review · Reviewer_d6hE · 2023-10-15

**Soundness:** 3 good
**Presentation:** 3 good
**Contribution:** 2 fair
**Rating:** 5
**Confidence:** 4

**Summary:**

Suite of Combinatorial Optimization benchmarks in JAX. Some experiments with an AC algorithm

**Strengths:**

New benchmarks are always good, especially CO, where fewer benchmarks are available.

**Weaknesses:**

No comparisons to other benchmarks or implementations. It is a sympathetic and perhaps substantial effort, but lacks elements that would achieve wide adaptations. The software engineering is there, the science is unclear.

There are Gym-JAX environments, and there are CO-Gym implementations (OR-Gym).

It appears that Jumanji does not follow the Gym interface. Stable Baselines algorithm are therefore not a drop in plugin.

Explain clearly the difference with a Gym interface. Why this choice?

Carrying around explicit state deviates from an RL principle, that the environment has the state, and not the agent.

Experimental validation with limited algorithms. No comparison to other benchmarks.

**Questions:**

What is the contribution of this paper?

Wouldn’t it make more sense to remain Gym-compliant in providing a Gym-JAX-CO implementation? This remains implicit, and is not explained.

Would a wrapper be possible for a Gym API? Could you use stable baselines unchanged?

---

> ### Author Response · Authors · 2023-11-15
>
> We thank the reviewer for their comments and feedback. We kindly provide answers to their questions and hope that the provided clarifications are sufficient for reassessing the recommendation for publication.
>
> > Carrying around explicit state deviates from an RL principle, that the environment has the state, and not the agent.
>
> We agree with the reviewer that carrying the state explicitly could make some algorithmic implementations deviate from the MDP principle. This is because implementing simulators in JAX requires functions to be pure, this means their computation graph must link inputs to outputs without any state variables. This paradigm is in line with other JAX-based RL environments like Brax and Gymnax. The RL formulation differentiates states (used by the environment) from observations (given to the agent). In the Jumanji API, methods return a tuple `(state, timestep)`. Therefore, the RL principle is conserved if one implements an environment loop that is responsible for giving the state to the environment and the timestep (which contains the observation) to the agent. An example of such a training pipeline is provided in `jumanji.training.agents.a2c.A2CAgent` and was used to obtain the learning curves from Figure 1.
>
> > What is the contribution of this paper?
>
> As opposed to demonstrating an algorithmic novelty, this work introduces a benchmark suite for RL research. Jumanji is an open-source and diverse suite of industry-inspired RL environments that are fast, flexible, and scalable. Within the COP community, recent works [Kwon et al., 2020; Hottung et al., 2022; Grinsztajn et al., 2023] have tackled problems such as generalization to bigger problem sizes and different problem distributions. Follow-up works could benefit from using Jumanji to train and evaluate their methods on a wide distribution of problems.
>
> As such, Jumanji is a tool for RL research, for which we provide baseline actor-critic agents as a point of reference for researchers to build upon.
>
> > Wouldn’t it make more sense to remain Gym-compliant in providing a Gym-JAX-CO implementation? This remains implicit, and is not explained.
>
> Gym environments are stateful in the sense that their methods are impure and thus cannot be compiled with JAX. This is the approach other JAX RL libraries like Gymnax have chosen, diverting from the Gym API to make the environments’ methods stateless. In Jumanji, we have chosen the `dm_env` API based on a timestep to make the function signatures lighter. This way, both reset and step methods return a state and a timestep. The former contains everything the environment needs to run the dynamics, theoretically, it is hidden from the agent. The timestep is a namedtuple that contains everything the agent sees, i.e. the observation, reward, discount, etc. This API avoids having 4 or 5 returned objects like one can find in Gym or Gymnax. It also prevents us from mixing environment data and agent observation like in Brax.
>
> > Would a wrapper be possible for a Gym API? Could you use stable baselines unchanged?
>
> A `JumanjiToGymWrapper` is already implemented via `from jumanji.wrappers import JumanjiToGymWrapper`. This wrapper converts a Jumanji environment to a `gym.Env` environment. The downside to this is that, although the step and reset functions can be jitted, the full environment loop (acting for multiple steps and training) cannot be jitted. By switching to a stateful paradigm (e.g. Gym), one loses the ability to fully train end-to-end on a device [Hessel et al., 2021]. We have amended appendix A.4 to precise the currently implemented wrappers in Jumanji, which include the gym wrapper.
>
> **References**:
> - Hessel et al. Podracer architectures for scalable reinforcement learning. 2021.
> - Kwon et al. POMO: Policy Optimization with Multiple Optima for Reinforcement Learning. NeurIPS (2020).
> - Hottung et al. Efficient Active Search for Combinatorial Optimization Problems. ICLR (2022).
> - Grinsztajn et al. Winner Takes It All: Training Performant RL Populations for Combinatorial Optimization NeurIPS (2023).

---

> > ### Comment · Reviewer_d6hE · 2023-11-22
> >
> > Thank you very much for your detailed response. In light of your response and the other reviews, I feel that my rating is accurate and leave it unchanged.

---

### Official Review · Reviewer_VTas · 2023-10-29

**Soundness:** 4 excellent
**Presentation:** 4 excellent
**Contribution:** 4 excellent
**Rating:** 8
**Confidence:** 4

**Summary:**

The authors propose Jumanji, a diverse set of accelerated environments written in JAX focused on NP-hard combinatorial optimization problems (COPs). Jumanji is fully open-source, fast, flexible, and scalable, covering 18 environments such as TSP (Travelling Salesman Problem). The authors also present A2C learning curves in these 18 environments to demonstrate end-to-end learning. Interestingly, Jumanji can tune the difficulties of the environments, showing that these environments can get exponentially more difficult to solve.

**Strengths:**

* Open-source accelerated environments in COPs: most of the accelerated environments are in robotics (e.g., NVIDIA's isaacgym or Google's brax), but I like the authors specific focus on NP-hard optimization problems.
* Optimal performance in some games: I like the authors added the reference optimal performance in some of the 18 environments.

**Weaknesses:**

I do not see any major weakness. One issue is that Figure 3 does not seem like a fair comparison with GPU. In particular TPU-v4s should be compared with A100s instead of RTX 2080 Super.

**Questions:**

I am curious why the authors chose A2C as the training algorithm instead of P

---

> ### Author Response · Authors · 2023-11-15
>
> We thank the reviewer for their feedback.
>
> > I do not see any major weakness. One issue is that Figure 3 does not seem like a fair comparison with GPU. In particular TPU-v4s should be compared with A100s instead of RTX 2080 Super.
>
> We agree with the reviewer that the performance obtained with A100s should be closer to that of TPUs, but the formers were not available at the time of running the experiments. We would like to clarify that the experiment illustrated in Figure 3 is meant to highlight the increase in throughput as one scales the available hardware, and not to showcase differences between TPUs and GPUs.
>
> > I am curious why the authors chose A2C as the training algorithm instead of P
>
> Indeed, we chose the A2C algorithm for our benchmark instead of, say, PPO or other more advanced algorithms. We purposely wanted to show how simple algorithms, when combined with neural network architectures that account for the symmetries of the considered problem, can achieve reasonable performances. This benchmark can then be used by researchers to build better-performing algorithms and use our A2C performance as a reference.

---

### Official Review · Reviewer_S6or · 2023-10-31

**Soundness:** 2 fair
**Presentation:** 3 good
**Contribution:** 2 fair
**Rating:** 6
**Confidence:** 4

**Summary:**

This paper presents a Jax based RL environment suite called Jumanji. The 18 environments focus on combinatorial optimization problems, designed to be fast, flexible, and scalable. They also provide an A2C benchmark and examples to motivate these problems.

**Strengths:**

- RL for many years has struggled with good environment code maintenance and support, and it is good to see this problem continue to be addressed
- The code base seems to be well designed and documented, the doc strings are generally informative and type hints are present.

**Weaknesses:**

- NP Hard optimization style problems have seen some interest in RL, but are not as common in literature, it would be beneficial to have more citations justifying their uses or explain more how common RL problems can be rethought into the COP formalism
- Having some sort of UML or diagram would be of great help to understanding the API.
- I don’t think random policy adds anything in Figure 2. It is expected that random does poorly and I’m not sure it adds much (given the trends of the curves, the impression of learning comes across)
- I’m not sure how much the y-axis labels matter in Figure 2 given how much clutter they add. A lot of these environments are not super common (and even in common Atari environments human normalised performance is increasingly common as a metric since the actual scores don’t mean much to most people). As long as they are all linear axes, and the optimal performance is there, all that matters is that the lines are going up (since this isn’t an algorithm paper, this figure is just showing things can learn in your environments).
- A plot like Fig 3(b) with number of TPUs vs. time to reach a certain performance could make a good figure (for the appendix at the very least)
- If CPU is not visible on the plot, I would just leave it off the labels and keep the text remark
- Although there are a lot of different environments implemented, it would be beneficial to have a point of comparison. As the authors note, there has been a fair amount of work in high performance environments already. Even if you can’t make a 1 to 1 comparison (because the environments are not the same), finding something of comparable complexity and having a figure in the appendix would help to ground the speedups.

**Questions:**

- How important is hardware flexibility? Are TPUs widely used outside google?
- Gamma is put in the MDP formalism of Jumanji. Although this can be seen both in and outside of the tuple, is there any explicit representation of it in the software? I.e. in the Jumanji environments, clearly all the other elements of the tuple are required to be defined for a functioning environment, but is the gamma represented?
- It would be beneficial to give more of an explanation of the state, just another sentence or so, explaining (perhaps with an example) what it is and contains. I assume it is a pytree (since the observation is), but is the key element required? Does step have to split the key necessarily if it doesn’t use it (small details like this could go in the appendix)?
- Environment version control is mentioned, but how often are changes made that increment this version? Version control is nice, but if there are hundreds of versions, it isn’t a panacea.
- Appendix C2 demonstrates weak (sometimes negative) scaling on CPU. Why is this the case? I would expect some speedup up to the 8 cores (assuming you are mapping across all cores, jax by default will just work with 1 (https://github.com/google/jax/issues/5022).
- Why does figure 3a start at 2^7 environments? The on many of the environments doesn’t seem as impressive as it could if this started at 2^0 perhaps
- Why is it called Jumanji?

---

> ### Author Response · Authors · 2023-11-15
> **Official Comment by Authors (Part 1)**
>
> We thank the reviewer for their helpful suggestions and we hope we have answered all their questions below.
>
> > NP Hard optimization style problems have seen some interest in RL, but are not as common in literature, it would be beneficial to have more citations justifying their uses or explain more how common RL problems can be rethought into the COP formalism
>
> We have adjusted the text in the "Combinatorial Optimization Problems" paragraph of the related work section to better emphasize how COPs can be tackled with RL. We included the works of [Kool et al., 2018; Hottung et al., 2022] which proposed a transformer architecture to efficiently treat COPs as RL episodes.
>
> > Having some sort of UML or diagram would be of great help to understanding the API.
>
> We thank the reviewer for the feedback and agree that a UML diagram would improve the clarity of the code base and provide useful insight for new developers working with Jumanji. We will provide this as part of the appendix and in the repository documentation.
>
> > I don’t think random policy adds anything in Figure 2. It is expected that random does poorly and I’m not sure it adds much (given the trends of the curves, the impression of learning comes across)
>
> Although a random policy is expected to perform poorly, we have decided to include this in our plots as we have repeatedly found that the random policy score is a useful lower bound for debugging new algorithms. An agent whose neural network parameterization and/or algorithm is incorrect could perform close to the random policy in expectation. Therefore, it helps to assess algorithm implementations and we think it will be useful to the community as well.
>
> > I’m not sure how much the y-axis labels matter in Figure 2 given how much clutter they add.
>
> We agree with the reviewer that the reward scales are generally arbitrary, and do not bring in itself more information than the relative improvements we observe during the training. However, in some environments (e.g. TSP or CVRP where the reward is the negative tour length, JSSP where it’s the negative makespan), they directly correspond to the underlying problem objective and align with previous work, so having it could make the comparison easier in general.
>
> > A plot like Fig 3(b) with number of TPUs vs. time to reach a certain performance could make a good figure (for the appendix at the very least)
>
> We thank the reviewer for their feedback and agree that this would be insightful to the reader. We have amended section C.2 of the appendix with such a table, and we hope it helps better understand the experiment.
>
> > Although there are a lot of different environments implemented, it would be beneficial to have a point of comparison. As the authors note, there has been a fair amount of work in high performance environments already. Even if you can’t make a 1 to 1 comparison (because the environments are not the same), finding something of comparable complexity and having a figure in the appendix would help to ground the speedups.
>
> Since we can’t compare the throughput of a Jumanji environment with a non-JAX version of it, we can at least demonstrate the difference in throughput between running the environment on a CPU (Figure 7b) and on a GPU (Figure 3a). We observe that running the environment on an accelerator brings orders of magnitude speed-ups compared to having the simulation on a CPU.
>
> > How important is hardware flexibility?
>
> As we are focusing on providing tooling that will be used by the greater research community we believe that hardware flexibility is a valuable attribute of any modern research library. Many researchers will be in a position where they only have access to CPUs, while others will want to utilize GPUs. Although many researchers will not have access to TPUs, we believe it is important for the library to also have the flexibility to be used by these proprietary architectures. Additionally, the flexibility of XLA means that JAX-based libraries like Jumanji are future-proofed to new advances in hardware.
>
> > Are TPUs widely used outside google?
>
> We acknowledge the reviewer's observation that many researchers do not have access to TPUs. However:
>
> - There is still a sizable portion of the RL community, in both academia and industry, using these devices outside of Google through the TRC program, as shown by this [publication list](https://sites.research.google/trc/publications).
> - Importantly, we highlight that Jumanji can be used with any XLA-compatible device (e.g. GPU). We emphasize that this library is not restricted to TPU users.

---

> ### Author Response · Authors · 2023-11-15
> **Official Comment by Authors (Part 2)**
>
> > Gamma is put in the MDP formalism of Jumanji. Although this can be seen both in and outside of the tuple, is there any explicit representation of it in the software? I.e. in the Jumanji environments, clearly all the other elements of the tuple are required to be defined for a functioning environment, but is the gamma represented?
>
> Gamma is usually specified as part of the problem (i.e. the MDP). However, most environment implementations (e.g. in Gym) do not include gamma. Therefore, we have adhered to the standard practice and have not included it. Moreover, all Jumanji environments have a limited horizon, which allows for undiscounted values (gamma = 1). Nonetheless, we mention gamma in Section 3.1 when properly defining the RL objective.
>
> > It would be beneficial to give more of an explanation of the state, just another sentence or so, explaining (perhaps with an example) what it is and contains. I assume it is a pytree (since the observation is), but is the key element required? Does step have to split the key necessarily if it doesn’t use it (small details like this could go in the appendix)?
>
> We thank the reviewer for their feedback and have accordingly updated the **State** paragraph of section 3.2 for better clarity. The reviewer is correct in that the state is a pytree. Additionally, the key is needed in order to handle stochasticity in wrappers like the `JumanjiToDMEnvWrapper`. If the environment is not stochastic, i.e. if it does not use the key, then it does not have to split it. On the contrary, it will split the key when any stochastic computation is needed within the step function.
>
> > Environment version control is mentioned, but how often are changes made that increment this version? Version control is nice, but if there are hundreds of versions, it isn’t a panacea.
>
> We agree and therefore aim to use the version control in the environments sparingly, only increasing the version after changes that affect the environment's behavior/performance. This policy is in line with that followed in Gymnasium (formerly, OpenAI Gym), where there are at most a handful of versions for each environment.
>
> > Appendix C2 demonstrates weak (sometimes negative) scaling on CPU. Why is this the case? I would expect some speedup up to the 8 cores (assuming you are mapping across all cores, jax by default will just work with 1 (https://github.com/google/jax/issues/5022).
>
> We emphasize that Fig 7.b displays the scaling of the environment performance on a 2-core CPU and not a TPU.
>
> > Why does figure 3a start at 2^7 environments? The on many of the environments doesn’t seem as impressive as it could if this started at 2^0 perhaps
>
> The higher the number of parallel environments, the greater the parallelization on hardware accelerators. As a reference, Brax [Freeman et al., 2021] trained with a batch size of 2048 and experimented with a number of parallel environments ranging from $2^7$ to $2^{16}$.
>
> > Why is it called Jumanji?
>
> In this context, 'Jumanji' is a metaphorical reference to the jungle, representing the complex and dynamic nature of scalable RL environments in JAX, much like the unpredictable jungle challenges in the Jumanji series.
>
> **References**:
> - Kool et al. Attention, learn to solve routing problems! ICLR (2018).
> - Freeman et al. Brax - A Differentiable Physics Engine for Large Scale Rigid Body Simulation, 2021. URL: http://github.com/google/brax.
> - Hottung et al. Efficient Active Search for Combinatorial Optimization Problems. ICLR (2022).

---

> ### Comment · Reviewer_S6or · 2023-11-19
>
> I appreciate the authors responses, and they have sufficiently addressed many of my listed concerns. As such, I am upgrading my recommendation (5->6)

---

### Official Review · Reviewer_7YMa · 2023-11-07

**Soundness:** 3 good
**Presentation:** 4 excellent
**Contribution:** 3 good
**Rating:** 6
**Confidence:** 4

**Summary:**

Jumanji is a suite of scalable reinforcement learning environments designed for RL research with industrial applications. It provides a collection of environments that are fast, flexible, and scalable, focusing on combinatorial problems and decision-making tasks. Jumanji leverages JAX and hardware accelerators to facilitate rapid research iteration and large-scale experiments. It stands out from existing RL environments by offering customizable initial state distributions and problem complexities and includes actor-critic baselines for benchmarking. The paper demonstrates Jumanji's high scalability and flexibility through experiments, positioning it as a tool to advance RL research.

**Strengths:**

Good paper and an important engineering contribution to an area of research in NP-hard combinatorial optimization problems (COPs). Solid design and software engineering work to make Jumanji modular, scalable, and fast and to fully unlock the power of hardware acceleration. The set of environments and tasks is complimentary in some sense to continuous control Jax-based training environments created by Google Brax team and will help to advance research in the area combinatorial problems and decision-making tasks.

**Weaknesses:**

A lack of a new research results and novel approaches. But it’s totally expected from such kind of more engineering oriented projects.

**Questions:**

What are the most important research challenges do you expect Jumanji will help to address?

---

> ### Author Response · Authors · 2023-11-15
>
> We thank the reviewer for their positive feedback and suggestions and hope that our answers clarify any concerns.
>
> > A lack of a new research results and novel approaches. But it’s totally expected from such kind of more engineering oriented projects.
>
> We agree with the reviewer that this work does not present new research results or approaches. Instead, our contribution focuses on utilizing advances in both hardware architecture and the high-performance computing library JAX to provide a test bed of hardware-accelerated environments. This contribution is impactful to the research community as it enables fast iteration of research and, due to the efficiency of our JAX environments, removes the need for large compute clusters to carry out meaningful RL research, as is the case with the majority of RL benchmarks.
>
> > What are the most important research challenges do you expect Jumanji will help to address?
>
> Within the COP community, recent works [Kwon et al., 2020; Hottung et al., 2022; Grinsztajn et al., 2023; Chalumeau et al., 2023] have tackled problems such as generalization to bigger problem sizes and different problem distributions. Follow-up works could benefit from using Jumanji to train and evaluate their methods on a wide distribution of problems. Additionally, we would like to invite researchers and engineers working on real-world applications to use the flexibility of Jumanji to tackle the very problems that hinder RL applications.
>
> **References**:
>
> - Kwon et al. POMO: Policy Optimization with Multiple Optima for Reinforcement Learning. NeurIPS (2020).
> - Hottung et al. Efficient Active Search for Combinatorial Optimization Problems. ICLR (2022).
> - Grinsztajn et al. Winner Takes It All: Training Performant RL Populations for Combinatorial Optimization NeurIPS (2023).
> - Chalumeau et al. Combinatorial Optimization with Policy Adaptation using Latent Space Search. NeurIPS (2023).

---

### Author Response · Authors · 2023-11-15
**General Answer to Reviewers**

We thank the reviewers for their helpful comments, feedback, and suggestions. In particular, we are pleased to see Jumanji recognized as a valuable contribution to the open-source RL community. Additionally, the reviewers found the manuscript well-written and the repository well-designed and documented.

We respond to each question and concern in detail for each reviewer independently. In general, the reviewers shared similar requests for clarification around the motivations behind the library and the contributions to the research community.

As hardware accelerators have become mainstream in academia and industry, there has been a growing need for JAX environments that allow just-in-time compilation and parallelization of simulations. The RL community needs JAX-based environments to speed up training and configurable simulators to scale the complexity of environments as methods improve. Jumanji removes the barrier of entry to students, research labs, and development teams by effectively utilizing hardware without the need for large compute clusters.

We thank the reviewers for raising these points among others as the modifications and additional results further strengthen our contributions. We believe that all raised points have been addressed and would be happy to discuss any remaining concerns that the reviewers may have.

---

### Meta-Review · Area_Chair_E1VB · 2023-12-06

**Metareview:**

This paper introduces a suite of hardware-accelerated RL environments, focused on combinatorial optimization problems. The reviewers largely agree that the Jumanji environment suite introduced in this work is an important contribution to RL as well as the field of combinatorial optimization, and can be expected to help advance the field.

**Justification For Why Not Higher Score:**

This work provides a useful suite of research environments featuring important combinatorial optimization problems that deserves to be widely published in a top venue like ICLR. However, from a methodological and even implementation standpoint compared to prior works, there is little novelty.

**Justification For Why Not Lower Score:**

This work presents a useful tool for many RL researchers, as well as researchers in combinatorial optimization. It has already been adopted by some researchers, and deserves to be published.

---

### Decision · Program_Chairs · 2024-01-16

Accept (poster)